# CAT-CAD: A Computer-Aided Diagnosis Tool for Cataplexy

Ilaria Bartolini *,†,‡  and Andrea Di Luzio †,‡

Department of Computer Science and Engineering (DISI), Alma Mater Studiorum, University of Bologna, 40126 Bologna, Italy; andrea.diluzio2@unibo.it
* Correspondence: ilaria.bartolini@unibo.it
† Current address: Viale del Risorgimento, 2-Bologna, Italy.
‡ These authors contributed equally to this work.

**Abstract:** Narcolepsy with cataplexy is a severe lifelong disorder characterized, among others, by sudden loss of bilateral face muscle tone triggered by emotions (cataplexy). A recent approach for the diagnosis of the disease is based on a completely *manual* analysis of video recordings of patients undergoing emotional stimulation made *on-site* by medical specialists, looking for specific facial behavior motor phenomena. We present here the CAT-CAD tool for automatic detection of cataplexy symptoms, with the double aim of (1) supporting neurologists in the diagnosis/monitoring of the disease and (2) facilitating the experience of patients, allowing them to conduct video recordings at home. CAT-CAD includes a front-end medical interface (for the playback/inspection of patient recordings and the retrieval of videos relevant to the one currently played) and a back-end AI-based video analyzer (able to automatically detect the presence of disease symptoms in the patient recording). Analysis of patients' videos for discovering disease symptoms is based on the detection of facial landmarks, and an alternative implementation of the video analyzer, exploiting deep-learning techniques, is introduced. Performance of both approaches is experimentally evaluated using a benchmark of real patients' recordings, demonstrating the effectiveness of the proposed solutions.

**Keywords:** automatic video content analysis; motor behavior patterns; mobile multimedia technologies; Artificial Intelligence for e-health

## 1. Introduction

Narcolepsy with cataplexy is a rare disorder mainly arising in young adults/children characterized by daytime sleepiness, sudden loss of muscle tone while awake triggered by emotional stimuli (cataplexy), hallucinations, sleep paralysis, and disturbed nocturnal sleep [1].

The current approach for the diagnosis of cataplexy exploits clinical interviews only [2]. However, such solution is prone to several pitfalls, ranging from the presence of different phenotypes, depending on age and disease, to the presence of symptoms in healthy subjects [3]. Recently, to overcome such limitations, a laboratory procedure for the diagnosis of cataplexy exploiting recognition and classification of abnormal motor behavior has been developed and validated [1]. This is based on a completely *manual* analysis of video recordings of patients undergoing emotional stimulation made *on-site* by medical specialists. However, currently no automatic technological solution exists, which is able to properly support neurologists in such a delicate task.

Few scientific studies have considered the video-polygraphic features of cataplexy in adult age and only recently the motor phenotype of childhood cataplexy has been described exclusively using video recordings of the attacks evoked by watching funny cartoons [1]. These studies showed that some distinctive elements of cataplexy, called *motor behavior patterns*, are present in the context of the physiological response to the laughter. Such patterns are particularly evident at the level of the *facial expression changes*. Other motor phenomena, like arm/leg falling, may also characterize cataplexy attacks, but some

patients might not be affected by such symptoms that, in any case, arise only at a later stage [3]. In particular, the three most recurrent motor phenomena displayed by patients affected by the disease are *ptosis* (a drooping or falling of the upper eyelid), *head-drop*, and *smile/mouth opening* [4]. According to this methodology, cataplexy is present if any of the three above symptoms is detected in the patient video. Such patterns are, however, still to be manually detected by neurologists by visually analyzing videos.

In this context, the development of a Computer-Aided Diagnosis (CAD) tool for the automatic recognition of disease symptoms could be extremely helpful, not only to help diagnosing the disease, but also in monitoring everyday activities in a non-invasive way to provide early warnings in the event of the insurgence of a crisis. Indeed, it is well known that the synergistic use of machine learning techniques can help in alleviating the burden of the medical specialist in analyzing patient data, thus improving diagnostic consistency and accuracy [5]. The requirements for such tool include:

- Automatic detection of cataplexy symptoms (ptosis, head-drop, mouth opening).
- Real-time video analysis, to provide instantaneous alerts to the patient treating doctor or parents.
- Video playback functionalities, to allow the neurologist to carefully investigate patient videos, possibly focusing only on sequences where symptoms were detected by automatic analysis. This has also to be complemented with the simultaneous analysis of automatically extracted symptom detections, so that results of video analysis must be displayed during the video playback.
- Similarity-based video retrieval, to allow comparing the video of the patient currently under analysis with videos of patients with similar behavioral patterns.

In this paper, we describe the CAT-CAD (Computer-Aided Diagnosis for CATaplexy) tool, which exploits Artificial Intelligence (AI) techniques for the automatic analysis of video content to support neurologists in the delicate task of the disease recognition/monitoring. CAT-CAD is the first tool to allow the automatic recognition of cataplexy symptoms based on the analysis of video content.

The contributions of this paper are (This paper represents a substantial extension of the preliminary work in [6], where we only presented the pattern-based recognition of ptosis):

- We introduce the CAT-CAD tool for the automatic recognition of cataplexy symptoms from patients' video recordings.
- We present the CAT-CAD GUI, showing its advanced functionalities for video playback and browsing/retrieval.
- We detail the pattern-based technique used to automatically analyze videos, able to recognize facial patterns.
- We introduce a novel approach for the detection of motor phenomena, based on convolutional neural networks.
- We provide results obtained from an extensive experimental evaluation to compare the performance of the two video analysis approaches, using a benchmark containing recordings from real patients.

The rest of the paper is as follows: Section 2 contains a description of related work, while Section 3 introduces CAT-CAD and details its front-end GUI for neurologists. In Section 4, we describe our approach for the automatic detection of ptosis, head-drop, and smile/mouth opening, and illustrate the setup used in the experiments. Section 5 reports results obtained on real data, while in Section 6 we discuss and interpret such results. Finally, in Section 7 we conclude, by drawing interesting directions for future studies. Useful background information on technologies exploited in CAT-CAD are provided in the appendix.

## 2. Related Work

To the best of our knowledge, to date the only study about the automatic recognition of cataplexy by exploiting patient video recordings is our proof of concept described in [6]. In any case, we believe it is useful to illustrate here some works that are related to ours, such as automatic detection of facial motor phenomena in other contexts (Section 2.1) and the use of deep-learning techniques for the analysis of images and/or videos in the domain of medical diagnosis (Section 2.2).

### 2.1. Detection of Motor Phenomena

Here, we report several studies that have been carried out to detect motor phenomena similar to the ones used to diagnose cataplexy, such as the detection of eyelid closure, head pose, or mouth opening for the automatic recognition of fatigue/drowsiness in vehicle drivers.

One of the works that recently addressed the problem is [7], which proposes a system for monitoring and detecting fatigue and distraction of bus drivers. The system consists of modules for head-shoulder detection, face detection, eye detection, eye openness estimation, drowsiness measure, percentage of eyelid closure estimation, and fatigue level classification. The paper describes an approach to estimate the level of eye openness based on spectral regression and a fusion algorithm to estimate the eye state based on adaptive integration on the multi-model detection of both eyes. The driver weariness and distraction are then classified using a measure of eye openness. A key feature of this system is that it can detect drowsiness on low resolution images of faces and eyes observed from an oblique viewing angle.

In [8], the authors exploit Histogram of Oriented Gradient (HOG) features [9] to build a framework for the real-time detection of driver drowsiness. In particular, the framework uses a cascaded AdaBoost classifier [10] based on Haar-like features to first localize the face within the current image, then the position of the eyes is detected using an active shape model algorithm. Adaptive HOG features are then extracted and fed to a Naïve Bayes classifier to detect the driver drowsiness. Results on a public video dataset (NTHU-DDD) show accuracy comparable to other state-of-the-art techniques with a frame rate of 24fps, making the system suitable for real-time analysis of driver videos.

Another study addressing the problem was presented in [11], where a driver-monitoring system is introduced. Methods for detecting both drowsiness and distraction are proposed: if the driver is looking ahead, drowsiness detection is performed, otherwise distraction detection is performed. The monitoring algorithm consists of steps for face detection, head orientation estimation, eye detection, eye state detection, drowsiness detection, and distraction detection. The driver's drowsiness level is measured as PERCLOS, which is the *PERcentage of eye CLOSure* time during a certain time interval. Similarly, the distraction level is measured as PERLOOK, which is the *PERcentage of time spent not LOOKing ahead* during a certain time interval.

A study worthy of mention is [12], where several facial features, including those of the eyes, mouth, and gaze, are integrated to measure the driver's vigilance level. These facial features are first located in the input video sequence, then the located facial features are tracked over the subsequent images (facial parameters are estimated during this phase). Finally, the estimated parametric values are collected and analyzed every fixed time interval to provide a real-time vigilance level of the driver.

Another work that combines PERCLOS and head movements tracking to detect driver inattention is [13]. The proposed system detects facial expressions, head movements, and the presence of some *visual cues* (that are known to reflect the driver's level of alertness) and combines all this information to detect user's drowsiness and distraction. This system can run in real time and to work with different people without a previous phase of training.

Finally, a study that combines mouth and eyes opening tracking to detect the hypo-vigilance state of a driver is [14]. Considering both mouth and eye opening, the authors solved the traditional challenge of detecting the fatigue of a driver wearing glasses. The

system locates the face region using the AdaBoost algorithm [10], then the positions of driver's eyes and mouth are located according to their geometric features. Finally, *Percentage And Time that Eyelids Cover the Pupils* (PATECP) and *Percentage And Time that Mouth Is Open* (PATMIO) are used to estimate the user's drowsiness.

Although discover of drowsiness and cataplexy symptoms are both based on the detection of eye/mouth/head movements, and are thus based on the analysis of similar facial features, their peculiar motor patterns are somewhat different: this prevents the "verbatim" use of techniques for drowsiness detection in the context of cataplexy diagnosis. Moreover, it must be remarked that according to the established procedure [15], patient videos are recorded in absence of sleepiness, thus drowsiness cannot be misidentified as ptosis (see also Section 4.3.2).

### 2.2. Deep Learning for Analysis of Medical Images/Videos

Recently, we all have witnessed the huge impact that deep learning is having on medical image analysis, thanks to the dramatically increased use of electronic medical records and diagnostic imaging [16–20]. In general, the main advantage of techniques based on machine learning is that they can automatically discover significant relationships within data without forcing the expert in the field to manually extract/detect relevant features. In particular, machine learning algorithms based on Convolutional Neural Networks are extensively used for medical image analysis (e.g., for classification, localization, detection, segmentation, and registration), emphasizing clinical aspects of the field [16–19], or for providing care services for impaired elder people [21].

Deep-learning approaches are however prone to several issues, among which the lack of labeled datasets for training, class imbalance in training data, and the inherent limit of explicability. Although there is consensus on the fact that more data improves accuracy of machine learning, the scarcity of data is even more critical in the medical context. Publicly available datasets are generally hard to find, and data labeled with an acceptable quality are even more rare (medical real datasets involve, on average, less than 100 patients [17]). Class imbalance consists of the skewing of images in the training data towards normal and non-pathological cases. Rare diseases are an extreme example of this and can be missed without adequate training examples, since it is hard to train a model without a sufficient number of examples. This effect can be softened by exploiting techniques of data augmentation, by synthetically generating additional training images of rare or abnormal data, although this could introduce overfitting [22]. Finally, one of the biggest issues with AI-based medical management methods is the one arising when a patient is misdiagnosed, and concerns legal and moral responsibility. With machine learning algorithms, this is accentuated by the difficulty to fully explain how they actually work, which makes nearly impossible to justify why a certain diagnosis was made [17].

The use of deep learning for the analysis of medical videos is still in its infancy, as only a few studies have exploited this technology for the detection of diseases using patients' videos [23,24]. However, the complexity of analyzing a video using a deep-learning approach consists of the introduction of the temporal dimension, which correlates consecutive frames. To the best of our knowledge, no previous study has analyzed this in the context of medical video analysis. Although three-dimensional CNNs [25] can in principle be used, such networks require an even larger number of training samples than a 2D CNN, due to the fact that a 3D network contains parameters whose number is orders of magnitude higher than a 2D network. This inevitably conduces to overfitting, given the limited amount of training data in medical applications [26]. To make things even worse, it should be remarked that in this case, data augmentation is extremely hard to achieve, due to the difficulty of producing additional synthetic videos [27], in particular for the case of pathological medical videos.

## 3. The CAT-CAD Tool

The overall architecture of CAT-CAD is depicted in Figure 1.

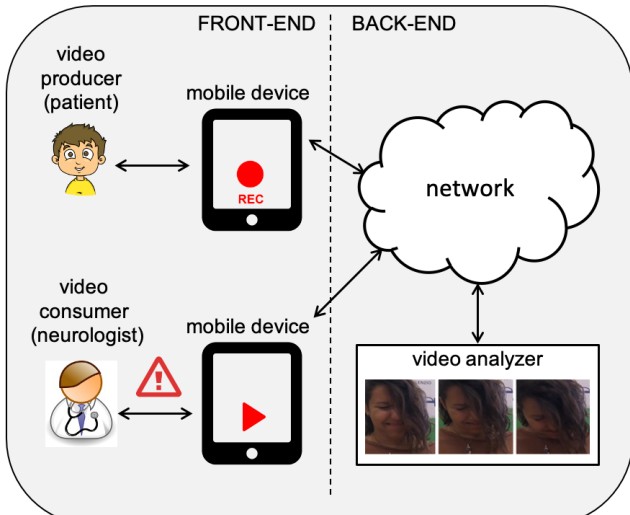

**Figure 1.** Architecture of the CAT-CAD (Computer-Aided Diagnosis for CATaplexy) tool.

In details, two actors are involved in the use of CAT-CAD:

1.  Patients play the role of "video producers", when videos are recorded (by exploiting a mobile device, such as an iPad/iPhone) showing their face during the vision of funny movies designed to evoke the laughter, thus providing emotional stimulation.
2.  Neurologists are "video consumers", receiving the output of visual analysis performed on their patients, such as the automatic "alert" generation in case of positive classification of the disease symptoms.

Video analysis is performed by a back-end service and is built on top of SHIATSU, a general and extensible framework for video retrieval which is based on the (semi-)automatic hierarchical semantic annotation of videos exploiting the analysis of their visual content [28]. This gives us several advanced services ranging from video frame splitting, frame feature extraction, and feature-based retrieval to visualization, data persistence, and automatic annotation. We can use such services for free, to focus our efforts on medical aspects only, to guarantee building an advanced and complete medical instrument. Details on SHIATSU that are relevant for our work are detailed in Appendix A.1. The video analyzer performs real-time analysis of the patient video to detect the presence of disease symptoms. The first technique developed to perform video analysis is a *Pattern-Based* approach (Section 4.1), built on the automatic detection of facial features in video frames. We also introduce an alternative *Deep-Learning* approach (Section 4.2) that exploits three convolutional neural networks, each trained to detect a specific motor phenomenon.

Finally, the medical tool features an intuitive and user-friendly Graphical User Interface (GUI), the CAT-CAD front-end, supporting neurologists with:

1.  automatic recognition of motor behaviors patterns to detect cataplexy symptoms, and
2.  video recordings searching and browsing facilities.

Figure 2 presents the intuitive and easy-to-use GUI designed for video consumers.

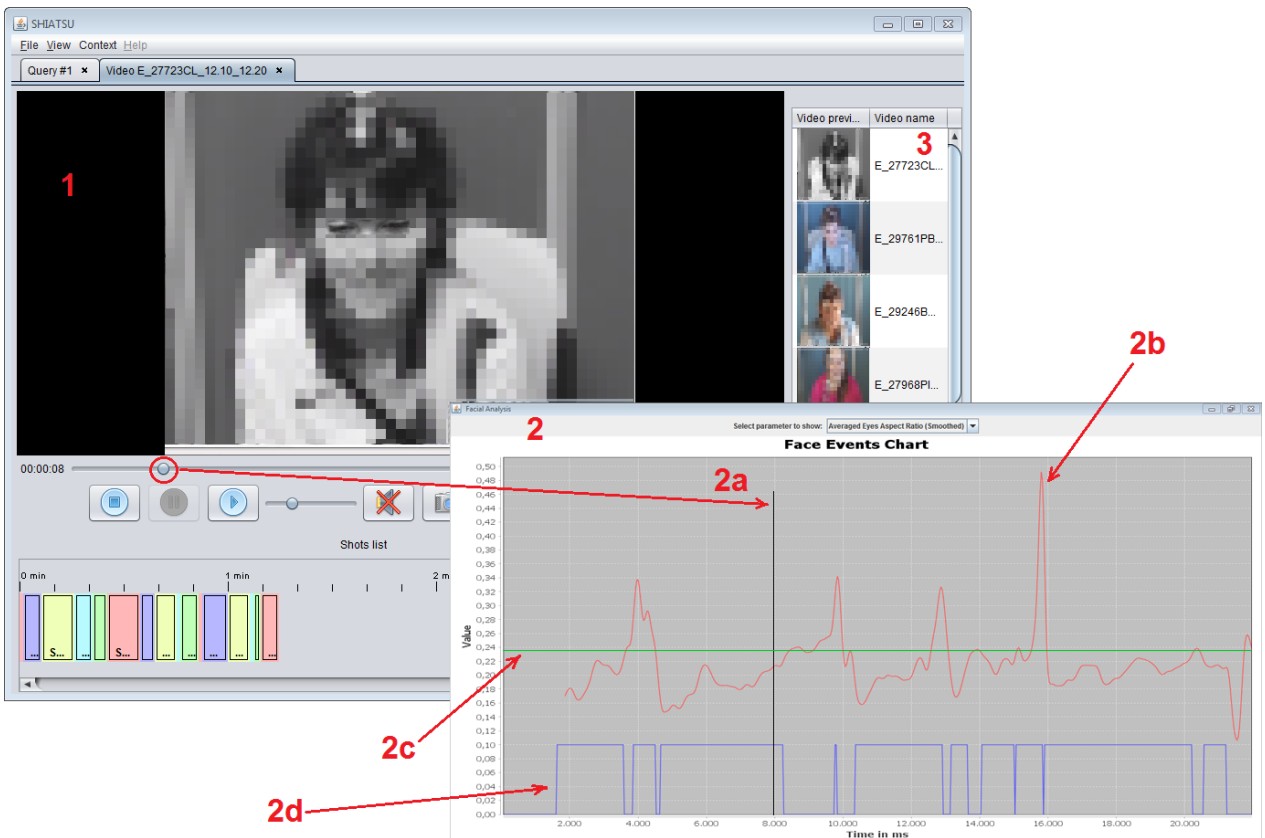

**Figure 2.** CAT-CAD front-end GUI: (**1**) Video playback area; (**2**) Face event chart panel, showing (**2a**) the video synchronization line, (**2b**) the currently selected facial feature, (**2c**) the reference value, and (**2d**) the motor phenomenon detection time series (the selected symptom is ptosis); (**3**) Similar videos panel.

The GUI allows the neurologists to study cataplexy presence in a patient by providing specific functionalities for inspecting both the original video recordings and the automatically computed features used to detect motor phenomena, displayed in the form of time series. Furthermore, neurologists have the possibility to search for video recordings of patients presenting features similar to a given one and to quickly identify possible correlations regarding a specific motor phenomenon in different patients.

The GUI basically consists of three different sections:

1. The central part represents the area where videos are played and contains playback control buttons; at the bottom of this area, the automatically segmented video shots are reported.
2. The external panel displays the face events chart, where the selected feature for the current patient is shown; graphs contained in this view include:
   (a) The vertical timeline, highlighting the current time value which is synchronized with the video playback.
   (b) The time series of the currently selected feature (in the figure, the Eye Aspect Ratio (EAR), see Section 4.1.1).
   (c) The reference value for the displayed feature.
   (d) The binary time series that indicates whether the selected motor phenomenon is detected or not for the current patient.
3. The right vertical panel contains videos which are similar to the currently displayed one; similarity between two videos is asserted by comparing the percentage of frames where the selected motor phenomenon is detected, and videos are sorted for decreasing values of similarity.

The physical separation of the panel containing the graph of motor phenomena from the main playback window is designed to allow the simultaneous viewing of graphs belonging to different videos, or belonging to the same video in case neurologists want to simultaneously observe several parameters of the same patient.

When the neurologist user starts video play, the chart panel is opened. The synchronization of the vertical timeline (2a) with the video playback allows the neurologist to compare the automatically detected motor phenomenon (2d) with corresponding facial expressions displayed by the patient in the currently played video (As it will be clear in the following, the time series (2b) and reference (2c) graphs are only available if the video was analyzed using the pattern-based approach, since the deep-learning analyzer does not use facial landmarks (see Section 4.2)).

## 4. Methods

This section details the methodology used to analyze patient videos to detect the presence/absence of cataplexy symptoms. In details:

- The Pattern-Based approach is presented in Section 4.1.
- The Deep-Learning approach is detailed in Section 4.2.
- Section 4.3 describes the setup used in the experiments.

### 4.1. Video Analyzer Back-End: Pattern-Based Approach

The first video analyzer to be implemented in CAT-CAD for the detection of cataplexy motor phenomena exploits facial landmarks, as detected by OpenFace (see Appendix A.2).

The first step of the pattern characterization process consists of detecting and extracting patients' facial landmarks of interest from each video frame; this is necessary because it is safe to assume that different patients have different facial features.

Using OpenFace on each single video frame we are thus able to extract, for each video, a time series of multi-dimensional feature vectors, each vector characterizing the facial landmarks extracted from a single frame. In the following, we detail the features extracted for the detection of ptosis (Section 4.1.1), head-drop (Section 4.1.2), and mouth opening (Section 4.1.3).

Extracted features are then used by the classifier to detect the presence of motor phenomena. Although any classifier could be used here (like decision trees, support vector machines, or even CNNs), the solution we purport exploits thresholds that are learned by optimizing prediction performance on available frames (see Section 5.1). This approach has the advantage of exposing the information about threshold values to the neurologist: as seen in Section 3, the CAT-CAD tool can show the reference value in the Face Event Chart Panel, giving the possibility of changing such value for a specific patient.

To reduce the effect of noise usually present in the feature time series, for each feature we also derive a smoothed version by applying triangular approximation. The smoothed version of each point of the original time series, $f_j$, is computed as the weighted average, $\widehat{f_j}$, of its 4 most adjacent points, computed as:

$$\widehat{f_j} = \frac{f_{j-2} + 2f_{j-1} + 3f_j + 2f_{j+1} + f_{j+2}}{9} \tag{1}$$

with $j$ in $[3, N-2]$, where $N$ is the number of frames in the video (For other values of $j$ the derivation is similar and not detailed here). We will experimentally evaluate the improved accuracy of smoothed features in Section 5.1.

#### 4.1.1. Ptosis

Ptosis is a drooping or falling of the upper eyelid. This, however, should not be mistaken as a (regular) eye-blink. For this, ptosis is detected whenever eyes are closed for a period of time longer than a typical blink. This is performed by analyzing facial features on each video frame to detect whether eyes are open or closed: the time of each eyelid

drooping can be measured as the number of consecutive frames for which the eye closure is detected multiplied by the frame rate.

For each frame, a 12-dimensional feature vector is extracted, containing the $(x, y)$ coordinates of the six landmarks $(\mathbf{p}_1, \ldots, \mathbf{p}_6)$ that characterize the shape of the eye (see Figure 3 for an example).

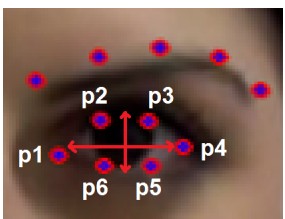

**Figure 3.** Landmarks of an open eye.

The *Eye Aspect Ratio* (EAR) is defined as the ratio of the eye *height* to the eye *width* [29], i.e.,:

$$EAR = \frac{\|\mathbf{p}_2 - \mathbf{p}_6\| + \|\mathbf{p}_3 - \mathbf{p}_5\|}{2\|\mathbf{p}_1 - \mathbf{p}_4\|} \tag{2}$$

where $\|\mathbf{x}\|$ denotes the norm of vector $\mathbf{x}$.

The semantics of EAR are as follows: when an eye is closing, EAR approaches zero, whereas when the eye is completely open, EAR attains its maximum value (which varies from person to person). EAR is partially invariant to head pose and fully invariant to uniform image scaling and in-place face rotation [29]. For a more stable characterization, the EAR value of the $j$-th frame, $EAR_j$, is assessed as the average of the left and right eye EAR values for that frame.

To precisely characterize the facial features of each patient under normal conditions, we performed the EAR extraction process on the video recordings at both base conditions (termed *baseline*) and while the patient was undergoing emotional stimulation. In this way, by measuring, for example, the median value ($\overline{EAR}$) of EAR values for the baseline video recordings, we obtain the characterization of the patient open eyes (because we can safely assume that during baseline recordings, patient eyes are normally open [30]).

Given the $j$-th frame of the recorded stimulation video, we consider the eyes as closed if the (relative) difference between $\overline{EAR}$ and the EAR for current frame, $EAR_j$, is higher than a threshold $T_{EAR}$:

$$\frac{\overline{EAR} - EAR_j}{\overline{EAR}} > T_{EAR} \tag{3}$$

Since threshold $T_{EAR}$ is an important parameter that strongly influences the accuracy of the detector, its optimal value is experimentally evaluated (see Section 5.1.1).

Finally, we define the presence or absence of ptosis by measuring the length of the time series corresponding to a "long enough" sequence of frames with closed eyes.

We therefore define ptosis as present if the amount of time when eyes are closed exceeds the maximum duration of an eye-blink, which can be measured as about 400 ms [31]. This is reflected in Algorithm 1, which detects presence/absence of ptosis on a series of feature values.

In line 3 of Algorithm 1, the *EC* time series is used to determine if, at the $j$-th frame, eyes are closed. Then, at line 4, function $blinks(j, w_P, EC)$ checks whether, in the $w_P$ frames preceding $j$ ($j - w_P, j - w_P + 1, \ldots, j - 1$) at least one of the *EC* values is false, i.e., eyes were opened in at least one of the preceding $w_P$ frames. The window length $w_P$ clearly depends on the frame rate $f$ and can be computed as $0.4 \times f$.

---

**Algorithm 1** Ptosis Detection

---

**Input:** Ptosis feature time series, $\overline{EAR}$, $T_{EAR}$, $w_P$
**Output:** $P$              ▷ Ptosis time series

1: **for** $j \leftarrow 1, N$ **do**
2:   compute $EAR_j$ as in Equation (2)
3:   $EC_j \leftarrow \left( \dfrac{\overline{EAR} - EAR_j}{\overline{EAR}} > T_{EAR} \right)$
4:   $P_j \leftarrow \neg blinks(j, w_P, EC)$

---

Finally, it must be highlighted that during videos with emotional stimulation, it can be the case that landmark extraction fails: this happens, for example, due to head-drops, which can be considered to be part of the ptosis (this is generally true, according to visual observations made on patients). To reflect this particular case, if we are unable to compute the value of $EAR_j$ at line 2 of Algorithm 1, the ptosis value is copied from the previous frame, i.e., $P_j \leftarrow P_{j-1}$.

4.1.2. Head-Drop

For head-drop, the 8-*D* feature vector extracted from each frame includes the $(x, y)$ coordinates of the two landmarks characterizing the external corner of each eye and the one that is immediately below the tip of the nose (see Figure 4a) and the rotation (in radiants) of the head around $X$ and $Z$ (see Figure 4b).

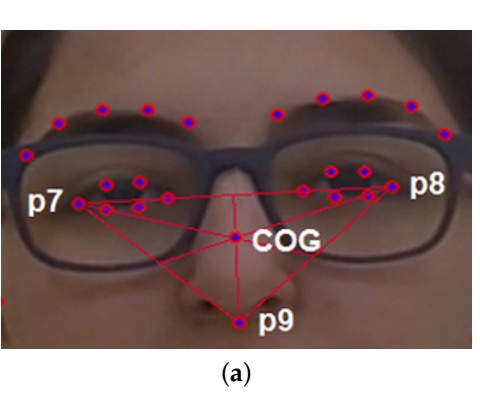

(**a**)

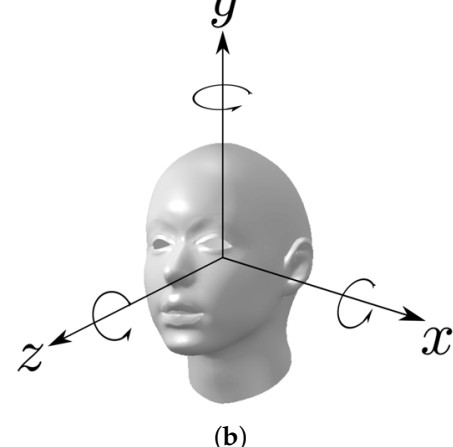

(**b**)

**Figure 4.** Position of the three facial landmarks considered and of the center of gravity computed from them (**a**) and head rotation around $X$, $Y$, and $Z$ axes (**b**).

We will analyze variations in position of the three facial landmarks as a whole, therefore it is useful to consider the center of gravity instead of the individual points. The *Center of Gravity* (CoG) of the three selected landmarks (see Figure 4a) is defined as:

$$CoG = \frac{\mathbf{p_7} + \mathbf{p_8} + \mathbf{p_9}}{3} \tag{4}$$

To measure the head-drop, we can use the trend over time of the $y$ coordinate of the center of gravity: this is correct since patients are always seated during the video recordings. In this way, we can avoid searching for cases when the values change because the patient has changed her/his position within the frame (that is, s/he stood up or seated).

As for the case of ptosis, by measuring the median value ($\overline{CoG}$) of $CoG$ values for the baseline video recordings, we obtain the position of the considered patient's face under normal condition.

Given the $j$-th frame of the recorded stimulation video, we consider a head-drop as present one of the following conditions is met for at least 300 ms (such value was experimentally defined by neurologists):

1.  The head moves down, i.e., the (relative) value of the CoG $y$ coordinate increases too much (as we measure $y$ coordinates from the top of the frame downward).

$$\frac{CoG_j.y}{\overline{CoG}.y} > T_{CoG}^{down} \tag{5}$$

2.  The head moves up or backward, i.e., the (relative) value of the CoG $y$ coordinate decreases too much.

$$\frac{CoG_j.y}{\overline{CoG}.y} < T_{CoG}^{up} \tag{6}$$

3.  Excessive head rotation around the $X$ axis.

$$HR_j.x > T_{ROT} \tag{7}$$

4.  Excessive head rotation around the $Z$ axis.

$$HR_j.z > T_{ROT} \tag{8}$$

As for ptosis, the optimal value for the three thresholds ($T_{CoG}^{down}$, $T_{CoG}^{up}$, and $T_{ROT}$) will be derived experimentally (see Section 5.1.2). Algorithm 2 can detect the presence/absence of head-drop using a time series of feature values.

---

**Algorithm 2** Head-Drop Detection

---

**Input:** Head-Drop feature time series, $\overline{CoG}$, $HR$, $T_{CoG}^{down}$, $T_{CoG}^{up}$, $T_{ROT}$, $w_{HD}$
**Output:** $HD$               ▷ Head-Drop time series
 1: **for** $j \leftarrow 1, N$ **do**
 2:      compute $CoG_j$ as in Equation (4)
 3:      $hd_j \leftarrow \left(\frac{CoG_j.y}{\overline{CoG}.y} > T_{CoG}^{down}\right) \vee \left(\frac{CoG_j.y}{\overline{CoG}.y} < T_{CoG}^{up}\right) \vee (HR_j.x > T_{ROT}) \vee (HR_j.z > T_{ROT})$
      ▷ Head-Drop is probably present at frame $j$
 4:      $HD_j \leftarrow long\_enough(j, w_{HD}, HDC)$

---

At line 4, the Boolean function $long\_enough(j, w_{HD}, HDC)$ checks whether the $j$-th frame is part of a sequence, lasting at least $w_{HD}$ frames, where the head-drop is always recognized as present. Again, $w_{HD}$ can be computed as $0.3 \times f$. Finally, as for ptosis, if the landmark extraction fails, the head-drop tag is copied from the previous frame, i.e., $HD_j \leftarrow HD_{j-1}$.

### 4.1.3. Smile/Mouth Opening

For the third motor phenomenon, mouth opening, an 8-$D$ feature vector is extracted for each frame, containing $(x, y)$ coordinates of the four landmarks that characterize the shape of the mouth (see Figure 5).

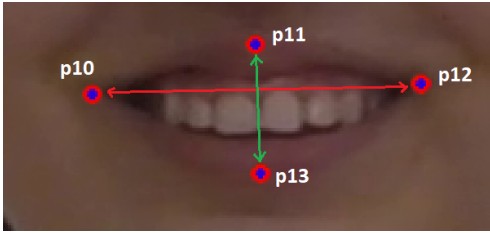

**Figure 5.** Example of landmarks of an open mouth.

The *Mouth Aspect Ratio* (MAR) can be defined as the ratio of the mouth *width* to the mouth *height*, i.e.,:

$$MAR = \frac{\|\mathbf{p}_{10} - \mathbf{p}_{12}\|}{\|\mathbf{p}_{11} - \mathbf{p}_{13}\|} \tag{9}$$

When the mouth is closed, MAR attains its maximum value (which varies from person to person), while if the mouth is completely open, MAR reaches his lowest value; intermediate values characterize various types of smile. Like EAR, MAR is partially invariant to head pose and fully invariant to uniform image scaling and in-place face rotation.

Differently from the case of ptosis, we performed the MAR extraction process only on the video recordings while the patients were undergoing emotional stimulation. The reason for this is that measuring the median value of MAR values for the baseline video recordings is not useful, because, due the brevity of such recordings, it may happen that the patient never closes his mouth for the entire duration of the video. To normalize MAR values, we thus take into account the minimum and maximum MAR values in the considered time series, computing normalized MAR (NMAR) values as:

$$NMAR_j = \frac{MAR_j - \min_i\{MAR_i\}}{\max_i\{MAR_i\} - \min_i\{MAR_i\}} \tag{10}$$

Given the $j$-th frame of the recorded stimulation video, we consider the cataplectic mouth opening as present if the current normalized MAR, $NMAR_j$, is lower than a threshold $T_{MAR}$ (indicating that the patient is smiling widely or opening his mouth):

$$NMAR_j < T_{MAR} \tag{11}$$

Again, the value of $T_{MAR}$ will experimentally optimized (see Section 5.1.3). For this motor phenomenon, we did not consider a minimum time span, given the brevity of some episodes of cataplectic mouth opening. Finally, as for the other two motor phenomena, if the landmark extraction fails, the mouth opening tag is copied from the previous frame, i.e., $MO_j \leftarrow MO_{j-1}$. The mouth opening detection algorithm is shown in Algorithm 3.

---

**Algorithm 3** Mouth Opening Detection

---

**Input:** Mouth Opening feature time series, $T_{MAR}$
**Output:** $MO$                           ▷ Mouth Opening time series
 1: **for** $j \leftarrow 1, N$ **do**
 2:     compute $MAR_j$ as in Equation (9)

 3: **for** $j \leftarrow 1, N$ **do**
 4:     compute $NMAR_j$ as in Equation (10)
 5:     $MO_j \leftarrow (NMAR_j < T_{MAR})$

---

Finally, based on the recognition of the three above-mentioned motor phenomena, a cataplexy attack is detected as present at frame $j$ if at least one of the three patterns is present in the given frame. This integration strategy for the patterns was indicated by the neurologists as the most correct [32].

### 4.2. Video Analyzer Back-End: Deep-Learning Approach

The alternative back-end video analysis tool is based on convolutional neural networks (CNNs) (A brief explanation of CNNs is included in Appendix A.3).

The CNN architecture used in this work is based on the DeXpression Network [33], which achieves excellent performance in expression recognition, and on GoogleNet [34], which has demonstrated how to significantly reduce the computational cost of deep neural networks without penalizing performance. The CNN has been implemented using TensorFlow, which is briefly described in Appendix A.4.

We have trained three different networks, one for each motor phenomenon to be recognized. The three networks share the same architecture, but the learned weights are clearly different, due to the use of different training classes. It is clear that for this approach,

each frame is analyzed per se, and no considerations on frame sequences, such as duration of eyelids closing or of head-drop, can be extrapolated, contrary to the pattern-based approach.

For each video frame, the face of the patient is first detected by means of OpenFace and then cropped. The resulting image is converted to grayscale and downsized from its original $1920 \times 1080$ size to produce images of $320 \times 320$ pixels. Cropping of the images was necessary to provide the CNN with only face details (thus avoiding that the surrounding environment would distract the learning).

Our CNN architecture consists of three different types of blocks, as detailed in Table 1:

1. an *Input Block*, which performs image pre-processing,
2. a *Feature Extraction Block*, inspired by the architectural principles introduced by GoogleNet, which is repeated four times, and
3. an *Output Block*, which is used to produce the result class from the features extracted by previous layers.

**Table 1.** Proposed CNN architecture: for each layer, the output size is reported.

| Block | Layer | Output Size |
|---|---|---|
| Input Block | Data | $320 \times 320$ |
| | Convolution 1 | $64 \times 160 \times 160$ |
| | Pooling 1 | $64 \times 80 \times 80$ |
| | LRN 1 | $64 \times 80 \times 80$ |
| FeatEx Block 1 | Convolution 2a | $96 \times 80 \times 80$ |
| | Convolution 2b | $208 \times 80 \times 80$ |
| | Pooling 2a | $64 \times 80 \times 80$ |
| | Convolution 2c | $64 \times 80 \times 80$ |
| | Concat 2 | $272 \times 80 \times 80$ |
| | Pooling 2b | $272 \times 40 \times 40$ |
| FeatEx Block 2 | Convolution 3a | $96 \times 40 \times 40$ |
| | Convolution 3b | $208 \times 40 \times 40$ |
| | Pooling 3a | $272 \times 40 \times 40$ |
| | Convolution 3c | $64 \times 40 \times 40$ |
| | Concat 3 | $272 \times 40 \times 40$ |
| | Pooling 3b | $272 \times 20 \times 20$ |
| FeatEx Block 3 | Convolution 4a | $96 \times 20 \times 20$ |
| | Convolution 4b | $208 \times 20 \times 20$ |
| | Pooling 4a | $272 \times 20 \times 20$ |
| | Convolution 4c | $64 \times 20 \times 20$ |
| | Concat 4 | $272 \times 20 \times 20$ |
| | Pooling 4b | $272 \times 10 \times 10$ |
| FeatEx Block 4 | Convolution 5a | $96 \times 10 \times 10$ |
| | Convolution 5b | $208 \times 10 \times 10$ |
| | Pooling 5a | $272 \times 10 \times 10$ |
| | Convolution 5c | $64 \times 10 \times 10$ |
| | Concat 5 | $272 \times 10 \times 10$ |
| | Pooling 5b | $272 \times 4 \times 4$ |
| ine Output Block | Classifier | 1 |

### 4.2.1. Input Block

The Input Block (Figure 6) contains a first Convolution layer, which applies 64 different filters with size $7 \times 7$ to input images, then a Max Pooling layer that down-samples them and, finally, a Local Response Normalization layer that normalizes the images.

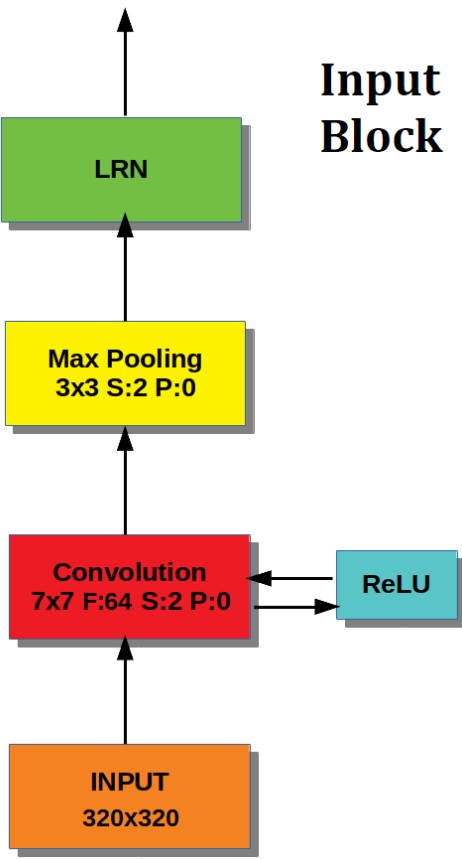

**Figure 6.** Input Block.

4.2.2. Feature Extraction Block

The second block type and the key element of our architecture is the Parallel Feature Extraction (*FeatEx*) Block [33], inspired by the architectural principles introduced by GoogleNet.

Each feature extraction block is formed by Convolutional, Max Pooling and ReLU layers organized in parallel (Figure 7).

- The left path provides a first Convolutional layer that uses a $1 \times 1$ filter to perform a bitwise analysis of the image, enhanced by a ReLU layer to create the desired sparseness [33], then the output is processed by a Convolutional layer with a filter of size $3 \times 3$, enhanced by a ReLU layer.
- The right path provides a Max Pooling layer (with $3 \times 3$ filter size) to reduce information followed by a (ReLU enhanced) Convolutional layer with $1 \times 1$ filter size.
- The output of the two paths represents face features at different scales and are concatenated together for a richer representation of the input image.
- Finally, the feature concatenation is input to a layer of Max Pooling (with $3 \times 3$ filter size or $7 \times 7$ in the last FeatEx block), which reduces the output size.

Unlike the network presented in [33], which used a cascade of two FeatEx blocks, we included four blocks, as in [34], to reduce the feature size for the last block down to $N \times 4 \times 4$.

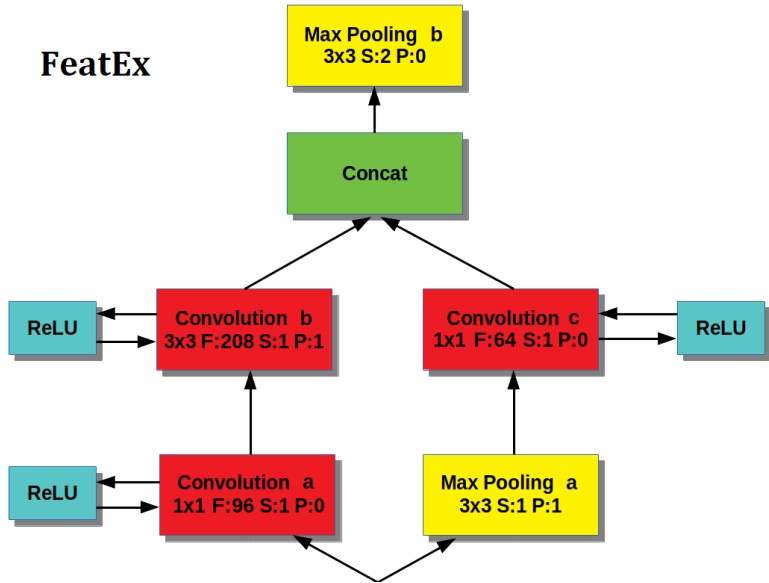

**Figure 7.** (Parallel) Feature Extraction Block.

4.2.3. Output Block

Finally, the features extracted in previous blocks are forwarded to a Fully Connected layer (the *Output Block*), which uses them to discern whether the image contains a specific motor phenomenon or not. In details, the features derived from previous blocks produce a probability for the two classes (presence or absence of the symptom): the class with higher probability is then output as the predicted one.

*4.3. Experimental Setup*

We here present the setup used in experiments performed to assess the performance of the two presented video analyzers.

- We first describe the population of subjects included in the study (Section 4.3.1).
- Then, we provide details on the video recordings used in the experiments (Section 4.3.2).
- Finally, we describe the performance measures used to assess the accuracy of the proposed approaches (Section 4.3.3).

All experiments were run on a 2C/4T @ 2.20 GHz CPU equipped with 4 GB of RAM, while experiments on CNN were performed using two NVIDIA Tesla K80 GPUs.

4.3.1. Population

The benchmark used for our experimental evaluation consists of a population of patients of the Outpatient Clinic for Narcolepsy of the University of Bologna, who were assessed for the presence of cataplexy by way of a neurophysiological and biological diagnosis [15].

The first (experimental) group of patients includes 14 subjects displaying symptoms of the disease. Details of this population are included in Table 2.

Examining videos recorded for these patients, all patients displayed symptoms of ptosis and smile/mouth opening during attacks, while only 4 patients (2, 3, 7 and 10) exhibit head-drop, and only patient 7 presents a significant fraction of frames where head-drop can be assessed.

Training of video analyzers has been performed using an inter-patient separation scheme, where patients have been randomly assigned to non-overlapping training and test sets, by respecting sex and age distribution. In particular, 11 patients have been included in the training set (thus, their entire videos have been used to train each analyzer), while the remaining 3 patients have been exploited to test the accuracy of the tool. Among the

patients exhibiting head-drop, 3 have been included in the training set and only one in the test set.

The second group includes 44 different subjects that show no sign of the disease. Among those, 14 patients have been selected as a control group to follow the same sex and age distribution of the experimental group.

**Table 2.** Description of cataplectic subjects (sex, age, average duration and number of cataplectic attacks in videos).

| Patient | Sex | Age | Avg. Duration (s) | No. of Attacks |
|---------|-----|-----|-------------------|----------------|
| 1 | F | 14 | 14.1 | 9 |
| 2 | F | 16 | 5.76 | 5 |
| 3 | F | 22 | 1.12 | 10 |
| 4 | M | 11 | 0.71 | 13 |
| 5 | M | 14 | 1.44 | 10 |
| 6 | M | 11 | 2.16 | 16 |
| 7 | M | 6 | 65.8 | 3 |
| 8 | M | 13 | 2.41 | 9 |
| 9 | M | 9 | 1.28 | 10 |
| 10 | M | 45 | 14.3 | 14 |
| 11 | F | 11 | 1.80 | 6 |
| 12 | F | 42 | 7.76 | 11 |
| 13 | F | 7 | 9.10 | 13 |
| 14 | M | 11 | 0.95 | 8 |

4.3.2. Video Recording and Labeling

Following the laboratory procedure detailed in [15], videos were recorded with the subject sitting in front of the camera for a time span of 10–40 min, long enough for the patient to feel at ease but not too long for sleepiness to set in. Video recordings included a baseline recording (up to 5 min), while for the rest of the recording subjects were watching videos [1]. All the videos were recorded at a resolution of 1920 × 1080 pixels with a constant framerate of 30 fps, using a very small and silent action camera (crucial to not distract patients), placed at about 70 cm from the patient while the monitor that shows funny videos has been placed at about 150 cm away from her/him. This allowed to film patients' faces with great precision. The choice of using of controlled high-quality videos was made to experimentally assess the accuracy of the CAT-CAD video analyzer without introducing noise. Indeed, testing the performance using low-fi videos that could be autonomously provided by patients using personally owned devices, such as smartphones or tablets, could result in worst results, which are however not completely imputable to the analyzer. We leave the performance evaluation of the CAT-CAD video analyzer with patients' produced videos as a subject for future investigation. To capture the head-drop motor phenomenon, the medium close-up shot was recorded.

Patients were asked to take a brief nap prior to video recordings to reduce sleepiness, and were asked again whether they felt sleepy at the end of the procedure; this was done to avoid the confounding effects of sleepiness on the observation of cataplexy. Videos of patients who reported sleepiness during the recordings were excluded from further analysis. Before starting each video recording, patients were interviewed to determine humor preference and asked to choose between various funny videos; these were selected to best stimulate each subject.

The overall recording procedure for the experimental and control groups generated an amount of ~370,000 frames (more than 3 h).

Finally, two experts in the field of movement disorders analyzed video recordings of patients undergoing emotional stimulation, classifying each motor phenomenon present in the video and specifying its beginning and end time, to create the ground truth to measure the accuracy of symptom detectors. Manual labeling of videos required over one month (clearly, only frames of the experimental group needed manual labeling).

Activities of video recording and labeling were performed by researchers of the Outpatient Clinic for Narcolepsy of the University of Bologna. The data used in this study consist of (i) the provided video frames and (ii) the corresponding labeling of motor phenomenon presence/absence.

### 4.3.3. Performance Measures

To objectively evaluate the performance of our analyzers, each frame can be labeled according to a confusion matrix as correctly and incorrectly recognized for each of the two classes available (in our case, motor phenomenon actually present or absent) [35]. Table 3 shows the four possible outcomes: $tn$ (true negative), $fn$ (false negative), $tp$ (true positive) and $fp$ (false positive).

**Table 3.** Confusion matrix for binary classification.

| Class/Recognized | As Present | As Absent |
| :---: | :---: | :---: |
| present | $tp$ | $fn$ |
| absent | $fp$ | $tn$ |

From the above confusion matrix, the performance measures used in our experiments are defined as follows.

**Recall/Sensitivity ($R$):** is defined as the fraction of the frames showing signs of the disease (positives) that are correctly identified. $R$ is therefore used to measure the accuracy of a technique in recognizing the presence of the disease:

$$R = \frac{tp}{tp + fn} \tag{12}$$

**Specificity ($S$):** is computed as the ratio of frames not showing the disease (negatives) that are correctly classified. $S$ thus expresses the ability of a technique to avoid false alarms (which can lead to expensive/invasive exams):

$$S = \frac{tn}{tn + fp} \tag{13}$$

**Precision ($P$):** is another popular metric, besides $R$ and $S$, which are the fundamental prevalence-independent statistics. $P$ is defined as the fraction of correct positively classified frames and assesses the predictive power of the classifier:

$$P = \frac{tp}{tp + fp} \tag{14}$$

**Accuracy ($A$):** measures the fraction of correct decisions, to assess the overall effectiveness of the algorithm:

$$A = \frac{tp + tn}{tp + fp + fn + tn} \tag{15}$$

**Balanced Score ($F_1$):** is a commonly used measure, combining $P$ and $R$ in a single metric computed as their harmonic mean:

$$F_1 = 2\frac{1}{\frac{1}{P} + \frac{1}{R}} = 2\frac{P \cdot R}{P + R} = \frac{2 \cdot tp}{2 \cdot tp + fp + fn} \tag{16}$$

All measures can be computed for any of the three motor phenomena, as well as for the *overall* cataplexy detection, which, we remind, is classified as present if any of the three motor phenomena is present. This has a direct impact on classification performance. Indeed, the following cases can be observed:

1. A $tp$ in the detection of any symptom will always correspond to a $tp$ in the overall detection.
2. Either a $fp$ or a $fn$ in the detection of a motor phenomenon can be transformed in a $tp$ in the disease detection, because another symptom has been detected. A $fn$ can be corrected by the detection of another motor phenomenon, while a $fp$ can be corrected by the fact that a cataplectic attack is nevertheless present, due to the actual presence of another motor phenomenon.
3. A $tn$ in the detection of a symptom can correspond to a $tp$, a $fp$, or a $fn$, according to a correct/incorrect detection of another symptom.

Thus, if we observe an improvement of the overall performance of any method over that a single symptom, this is due to the fact that a $fp$ or a $fn$ has been transformed into a $tp$ (case 2), while a decrease in accuracy is due to a $tn$ becoming a $fp$ or a $fn$ (case 3).

Finally, we note that when evaluating the performance in videos of non-cataplectic subjects, no positive cases are present, thus the performance of any classifier can be established by simply using specificity, since $P = 0$, $R$ and $F_1$ are undefined, and $A = S$. Moreover, the overall performance of each method (i.e., the one in detecting the disease) could only attain a specificity not higher than the one computed for any single motor phenomenon: in these cases, in fact, any $fp$ remains a $fp$, while it could be the case that a $tn$ becomes a $fp$.

Before discussing the results, it is necessary to point out that given the difficulty of the annotation work of the video recordings, a maximum tolerance of about 500 ms (corresponding to 15 frames, in our case) has been granted to the expert of the domain in labeling the beginning and ending of each motor phenomenon (or cataplectic episode). Such tolerance could not, however, be accounted during the evaluation phase, since, for every frame identified as the beginning or end of a cataplectic episode (or of a motor phenomenon), there was no way of knowing if and how much the domain specialists had benefited from the tolerance granted to them and, in case, whether this tolerance has been used to indicate the start and end frames of the motor phenomenon/cataplectic episode slightly earlier or slightly later than the real position in the video. For this reason, we evaluated techniques as if all the annotations of the ground truth were accurate to the frame level. This choice allows us to report objective results, instead of introducing new metrics that could produce results biased in favor of accuracy (for example, excluding from the calculation the 15 frames immediately before and after the start and end of each labeling).

## 5. Results

In this section, we report experimental results obtained using the two presented video analyzers. In particular:

- We first detail the procedure used to derive the optimal values for thresholds used by the pattern-based approach (Section 5.1).
- Then, we describe the data augmentation procedure necessary to train the CNNs for the deep-learning approach (Section 5.2).
- The overall performance of the two approaches is finally presented in Section 5.3.

### 5.1. Parameter Tuning

To determine the optimal value for the thresholds used in classifying the three motor phenomena in the pattern-based analyzer, we chose to exploit a "data-driven" approach, common for the tuning of machine learning algorithms, choosing the value that produces the best classifying performance on the test set [36]. The process involves fitting the model on a training dataset and making predictions on a test dataset, using different threshold values. The threshold that achieves the best evaluation metric is then adopted for the model

when making predictions on new data in the future. Receiver operating characteristics (ROC) graphs are widely used in medical decision making for diagnostic testing, since they provide a simple and intuitive way to evaluate performance of a classifier. A ROC graph is a 2-D graph with $S$ plotted on the X axis and $R$ plotted on the Y axis (We slightly abuse of notation here, since ROC graphs actually plot false positive rate, i.e., $1 - S$, on the X axis. Our choice avoids introducing another performance measure, without sacrificing the semantics of the graph). The best classifying performance, thus, can be assessed by choosing the threshold value closer to the optimal classifier performance (1,1). For this, we chose the threshold value that maximizes the harmonic mean of $R$ and $S$ measures: this represents a metric for imbalanced classification, seeking an equilibrium between the two measures [37].

### 5.1.1. Ptosis: Threshold $T_{EAR}$

Our first experiment aims at showing:

1.  how the default threshold $T_{EAR}$ in Equation (3) is assessed, and
2.  that the use of smoothed values (Equation (1)) improves performance.

Figure 8 shows $R/S$ values for different values of the threshold $T_{EAR}$. Clearly, when increasing the value of $T_{EAR}$, recall decreases, while specificity increases (because a larger number of frames would be recognized as positive). The optimal $T_{EAR}$ value is then computed as $T_{EAR} = 0.2$, which is the value that the software uses as default (in Figure 8, as well as the following, the optimal performance value is displayed as a vertical dashed line in R/S graphs and circled in ROC graphs). In any case, since the way to close the eyes during laughter may vary from person to person, a slider has been provided in the front-end GUI to adjust the threshold in case the neurologist deems it appropriate. As it will be shown in the experimental evaluation Section 5, this could lead to improve results with respect to the use of a unique threshold for all patients. Moreover, Figure 8 shows graphs for original ($EAR$) and smoothed ($\widehat{EAR}$) values of the EAR descriptors. It is clear that for all considered values of the threshold $T_{EAR}$, smoothed values of EAR achieve better performance, proving the higher accuracy of the proposed descriptors. Finally, we note that since videos were recorded with a frame rate of 30 frame/s, the window size for evaluating the presence of an eye-blink in Algorithm 1 is $w_P = 12$.

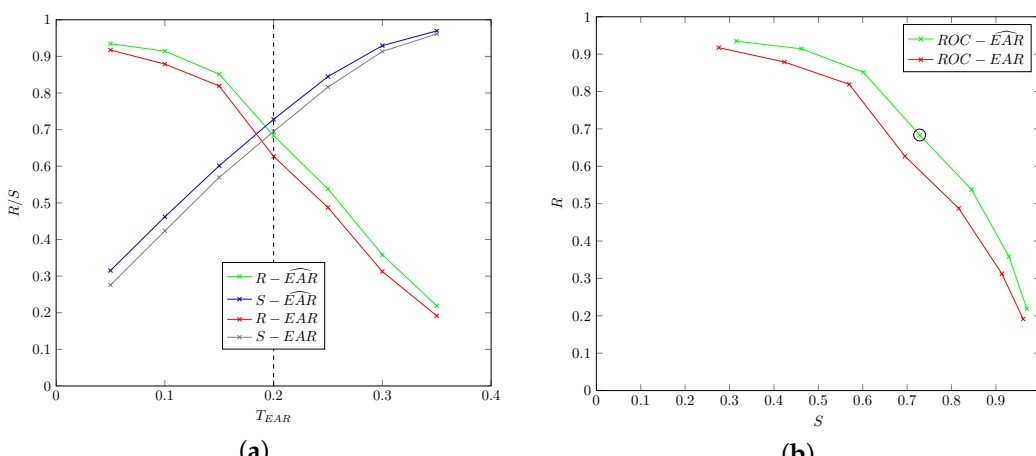

**Figure 8.** Recall ($R$)/Specificity ($S$) (**a**) and ROC (**b**) curves varying the threshold $T_{EAR}$ for original ($EAR$) and smoothed ($\widehat{EAR}$) time series. The optimal value is shown as a dotted line in (**a**) and circled in (**b**).

### 5.1.2. Head-Drop: Thresholds $T_{CoG}^{up}$, $T_{CoG}^{down}$, $T_{ROT}$

The goal of the second experiment we performed was to show:

1.  how thresholds $T_{CoG}^{up}$, $T_{CoG}^{down}$, and $T_{ROT}$ in Equations (5)–(8) are assessed, and
2.  that the use of smoothed values does not improve accuracy for the considered data.

Figures 9 and 10 show $R/S$ values for different values of thresholds $T_{CoG}^{up}$ and $T_{CoG}^{down}$. The optimal values are obtained for $T_{CoG}^{up} = 1.18$ and $T_{CoG}^{down} = 0.85$, respectively, and such values that will be used in all the subsequent experiments. Figures 9 and 10 also show graphs for original ($CoG$) and smoothed ($\widehat{CoG}$) values of the CoG descriptors: differences are minimal and are not really appreciable from the graphs (in fact, lines for original and smoothed values are almost perfectly overlapping).

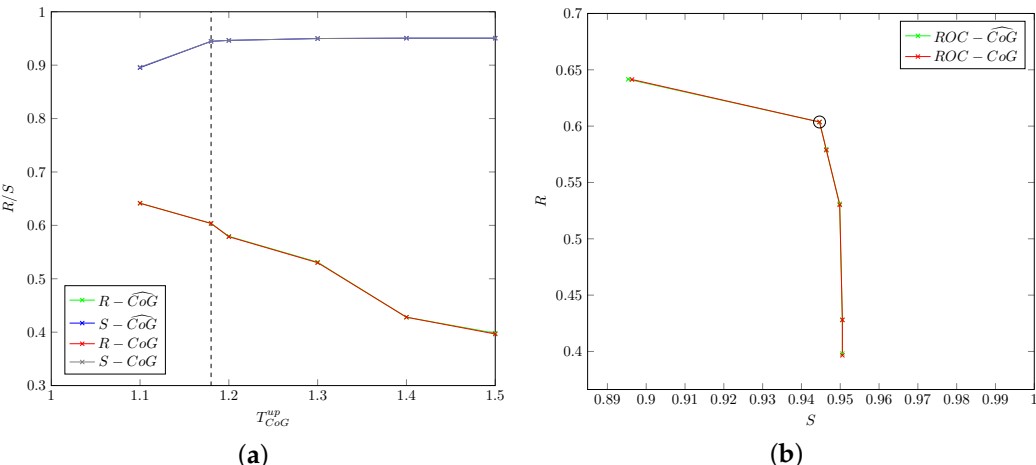

**Figure 9.** Recall ($R$)/Specificity ($S$) (**a**) and ROC (**b**) curves varying the threshold $T_{CoG}^{up}$ for original ($CoG$) and smoothed ($\widehat{CoG}$) time series. The optimal value is shown as a dotted line in (**a**) and circled in (**b**).

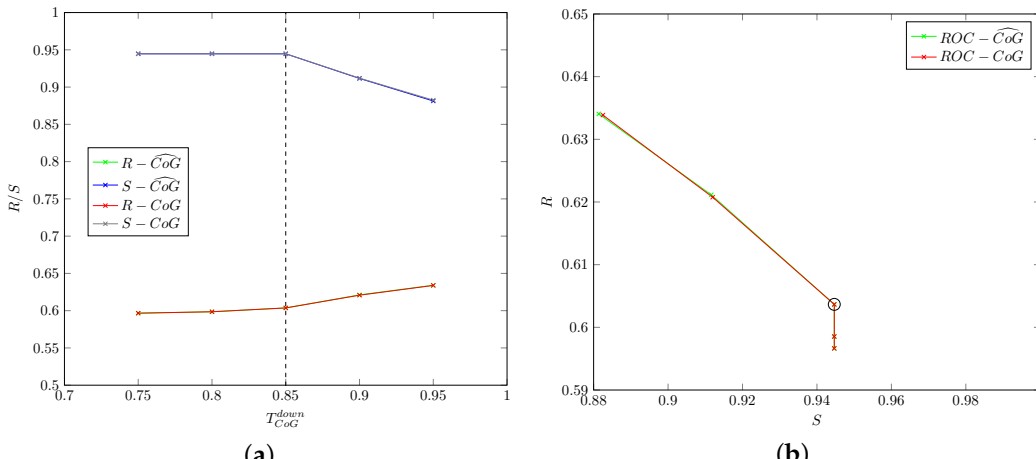

**Figure 10.** Recall ($R$)/Specificity ($S$) (**a**) and ROC (**b**) curves varying the threshold $T_{CoG}^{down}$ for original ($CoG$) and smoothed ($\widehat{CoG}$) time series. The optimal value is shown as a dotted line in (**a**) and circled in (**b**).

Finally, Figure 11 shows $R/S$ values when varying the threshold $T_{ROT}$. The optimal $R/S$ values are obtained for $T_{ROT} = 0.39$ (values of head rotation angle were not considered for smoothing, so the figure only includes original values).

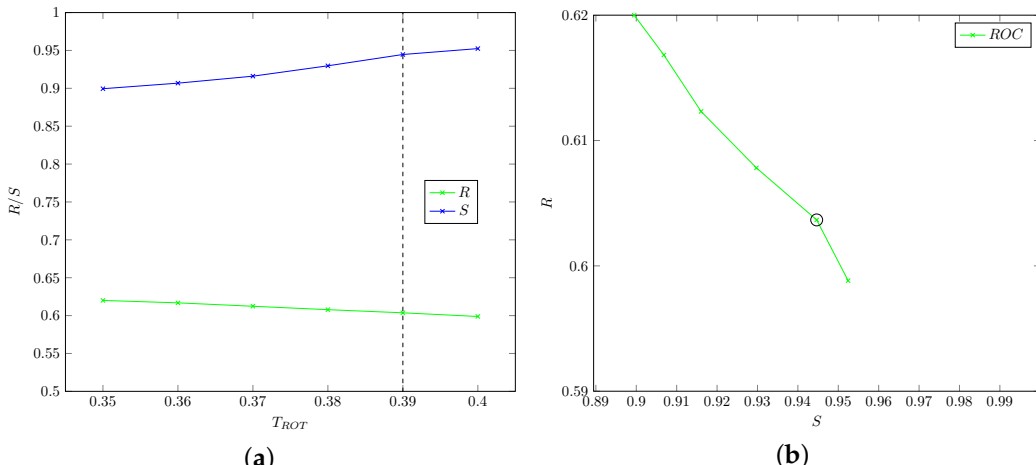

**Figure 11.** Recall (*R*)/Specificity (*S*) (**a**) and ROC (**b**) curves varying the threshold $T_{ROT}$. The optimal value is shown as a dotted line in (**a**) and circled in (**b**).

Again, frame rate at 30 frame/s led to assuming a window size for evaluating the presence of a head-drop in Algorithm 2 as $w_{HD} = 9$.

### 5.1.3. Mouth Open: Threshold $T_{MAR}$

The last parameter to be tuned is the $T_{MAR}$ threshold. Figure 12 shows $R/S$ values for different values of $T_{MAR}$ and for $MAR$ and $\widehat{MAR}$ time series. The optimal value for $T_{MAR}$ is 0.4 and it is also clear that smoothed values attain superior accuracy.

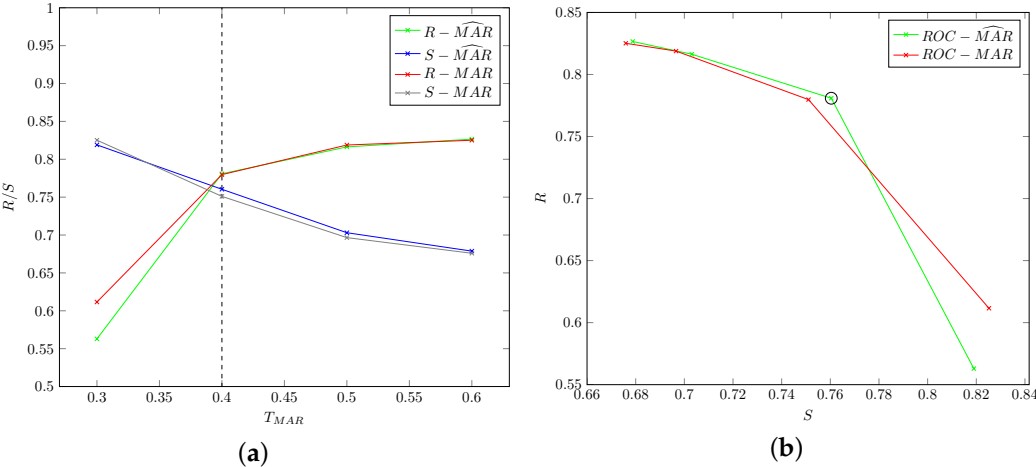

**Figure 12.** Recall (*R*)/Specificity (*S*) (**a**) and ROC (**b**) curves varying the threshold $T_{MAR}$ for original ($MAR$) and smoothed ($\widehat{MAR}$) time series. The optimal value is shown as a dotted line in (**a**) and circled in (**b**).

### 5.2. Data Augmentation Procedure

We detail here how data augmentation was performed for the deep-learning approach.

As noted before, in medical datasets for disease detection, it is common that the number of images in the training data is skewed towards non-pathological (negative) cases; a non-adequate number of positive cases can however lead the trained CNN to oversee symptoms of rare diseases [22]. For this, when building the datasets used to train the CNNs, we discarded most of the negative examples to have a balanced number of positive and negative examples [37].

Data augmentation was performed by adding, to each training set frame, seven additional images by performing:

- 3 *rotations* with a random angle between $-45°$ and $+45°$,

- 3 *translations* with a random shift between −50 and 50 pixels, and
- 1 *horizontal flipping*.

Figure 13 shows examples of the changes made to an image during the data augmentation phase.

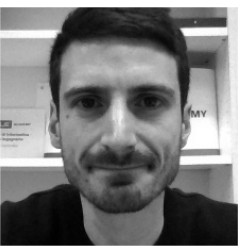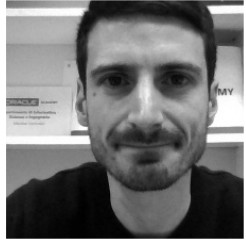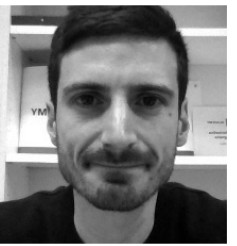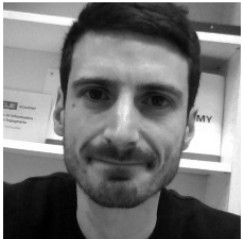

**Figure 13.** Examples of data augmentation. From (**left**) to (**right**): original (cropped) frame, translated, flipped, and rotated images.

Details of the training sets are as follows:

**Ptosis:** 191140 labeled images.

**Head-Drop:** 61216 labeled images.

**Mouth Opening:** 108196 labeled images.

The total time spent on building the three training sets was about four days.

*5.3. Overall Performance*

To compare the performance of the two alternative video analyzers, tables presenting results will show in boldface the best value obtained in any of the five considered performance measures.

5.3.1. Pattern-Based Approach

Table 4 reports the performance of the proposed pattern-based classification tool: overall figures values were obtained by averaging individual values weighted by the length of the considered recordings.

**Table 4.** Performance of the proposed pattern-based approach for cataplectic patients.

| Motor Phenomenon | *R* | *S* | *P* | *A* | $F_1$ |
| :---: | :---: | :---: | :---: | :---: | :---: |
| ptosis | **0.72** | **0.84** | **0.82** | **0.78** | **0.77** |
| mouth opening | **0.78** | 0.76 | 0.75 | **0.76** | **0.77** |
| head-drop | 0.60 | **0.94** | **0.89** | 0.77 | 0.72 |
| overall | **0.75** | **0.79** | **0.79** | **0.77** | **0.77** |

We highlight the fact that results for *R* and *S* reported in Table 4 are slightly better than those shown in Figure 8, thanks to the fact that for each patient we used the ptosis threshold calibrated by the neurologist in place of the default value.

Table 5 summarizes the performance of the video analyzer when considering videos of non-cataplectic subjects. Results are quite good, with an overall specificity at 98%.

**Table 5.** Specificity of the proposed pattern-based approach for non-cataplectic subjects.

| Motor Phenomenon | S |
|---|---|
| ptosis | **0.99** |
| mouth opening | **0.98** |
| head-drop | **0.99** |
| overall | **0.98** |

Finally, we include a brief discussion about efficiency of the proposed technique. On the experimental setup (which is typical of a low-end machine), we were able to extract EAR, CoG and MAR descriptors in real time for each video frame. Clearly, this is the more time-consuming operation in Algorithms 1–3, thus it is proven that the whole process of automatic detection can be performed on-line during a single emotional stimulated video recording session.

5.3.2. Deep-Learning Approach

The three neural networks have been trained for 8 epochs each, for a total time of about 12 h. The tests for the generation of confusion matrices, on the other hand, took about an hour and a half on the same platform. Table 6 report the average performance of the three trained neural networks on individual videos: as before, average values were calculated with a weighted average based on the length of the considered recordings. Furthermore, as for the pattern recognition approach, cataplexy is detected as present if at least one of the three motor phenomena is present in the considered frame, i.e., if at least one of the three networks has classified the considered frame as positive.

**Table 6.** Performance of the proposed deep-learning approach for cataplectic patients.

| Motor Phenomenon | R | S | P | A | $F_1$ |
|---|---|---|---|---|---|
| ptosis | 0.71 | 0.67 | 0.68 | 0.69 | 0.70 |
| mouth opening | 0.72 | **0.81** | **0.79** | 0.76 | 0.75 |
| head-drop | **0.67** | 0.81 | 0.78 | 0.74 | 0.72 |
| overall | 0.70 | 0.74 | 0.73 | 0.72 | 0.71 |

Table 7 summarizes the performance of our deep-learning approach regarding videos containing subjects that show no sign of the disease.

**Table 7.** Specificity of the proposed deep-learning approach for non-cataplectic subjects.

| Motor Phenomenon | S |
|---|---|
| ptosis | 0.83 |
| mouth opening | 0.83 |
| head-drop | 0.81 |
| overall | 0.66 |

Regarding efficiency, our current implementation of the deep-learning approach is only capable of obtaining a throughput of 18.5 frame/s, thus being unable to attain real-time performance (recall that the frame rate of videos is 30 frame/s). The reason for this measure is the following: when analyzing a single frame, about 50% of the time is spent in detecting the position of the patient face, about 25% in cropping the image (retaining only the face), and 25% for the classification of the frame by the three neural networks.

The bottleneck of the whole computation is clearly the face detection phase, which we implemented using the OpenFace library, instead of using other faster methods (such as the well-known Haar-Cascade filter [38]). This choice was carried out starting from the consideration that quicker filters often fail to identify the face within the image, especially

in videos with excessive head movement, which is the common case for cataplectic subjects. On the other hand, we believe that this bottleneck could be eliminated by using an additional GPU devoted to the face detection task (which unfortunately was not available in our experimental environment); this is obviously an issue we plan to investigate in the future.

## 6. Discussion

The comparison of results presented in the previous sections led us to draw the following considerations:

- The pattern-based approach led to significantly superior results with respect to its deep-learning counterpart. In particular, for cataplectic subjects the former attains the best performance in 85% of the metrics (17 out of $4 \times 5 = 20$ performance measures).
- When considering specific motor phenomena, the pattern-based approach consistently outperforms the deep-learning approach in detecting ptosis, while the latter sports superior measures only for specificity and precision in detecting mouth opening and for recall in head-drop detection.
- The superior specificity of the pattern-based technique is confirmed in non-cataplectic subjects.

A possible explanation for the inferior performance of the deep-learning approach is the fact that such approach cannot discriminate between quick and long eye blinks/head-drops, because each frame is analyzed individually by the CNN. It is, therefore, likely that the higher number of false positives is because the CNN wrongly detects "regular" eye blinks or head movements as ptosis or head-drop.

For the case of non-cataplectic subjects, it is interesting to note that the performance of the deep-learning approach for the overall detection of cataplexy is sensibly worse than those attained for the single motor phenomena. For such patients, false positives for ptosis, head-drop, and mouth opening are present in different frames. Indeed, due to the absence of positive cases, the set of false positive frames for the overall cataplexy coincides with the union of frames wrongly classified by any specific motor phenomenon detector, and an error in detecting a motor phenomenon cannot be "corrected" by a correct detection from a different symptom (as it could be the case for frames where the cataplexy is present).

Regarding the relevance of the three motor phenomena embedded in CAT-CAD, ptosis, head-drop, and smile/mouth opening are the most useful in discerning cataplexy, since they appear with high occurrence during cataplexy (regarding pseudo-cataplexy) attacks [4]. Other features include tongue protrusion, facial jerks and grimaces, and trunk fall. As proven by experiments, CAT-CAD can automatically detect any of the three motor features in patient-provided videos. The neurologist can then concentrate only on those portions of videos containing detected motor phenomena to diagnose the presence of cataplexy. Moreover, it must be highlighted that all three motor phenomena considered in CAT-CAD appear during the "initial" phase of cataplectic attacks, which is followed by "falling" and "atonic" phases. For this, CAT-CAD could also be helpful as an "early warning" tool for patients' relatives and/or physicians that a cataplectic attack is under way, before the most severe parts of the attack take place.

Although the video analyzer of CAT-CAD is specifically tailored for cataplexy, the software modules exploited to detect the motor phenomena could, in principle, be useful in diagnosing other clinical conditions. For example, head-drops are associated with a wide range of epileptic syndromes, movement disorders, and sleep disorders (like chorea, tics, paroxysmal dyskinesias, and Sandifer syndrome) [39]. The generality of the CAT-CAD architecture, in particular of the underlying SHIATSU framework, could also be exploited to extend the range of used cognitive features, allowing it to diagnose other diseases; for example, analysis of advanced motor symptoms could be used to detect neurodegenerative diseases (Parkinson, Alzheimer's, Amyotrophic Lateral Sclerosis, etc.) [24].

## 7. Conclusions

In this paper, we introduced CAT-CAD, a novel AI-based medical tool for the detection and monitoring of cataplexy based on automatic analysis of patient videos. CAT-CAD includes a GUI for video playback, searching, and retrieval, based on the SHIATSU software library. Automatic analysis of video recordings made on emotionally stimulated patients is carried out by exploiting results obtained from video classification. Our study focused on ptosis, head-drop, and smile/mouth opening, three of most recurrent cataplexy motor behavioral pattern. Two different approaches are introduced for the detection of disease symptoms: the Pattern-Based approach is based on analysis of facial features, using the OpenFace framework, while the Deep-Learning approach uses Convolutional Neural Networks, as implemented by Tensorflow. An extensive comparative experimental evaluation conducted on a benchmark of real patient recordings demonstrated the accuracy of the proposed tool.

When comparing the effectiveness of the two video analyzers we introduced to detect cataplexy symptoms, the pattern-based approach achieves superior performance. One of the possible explanations for the inferior detection accuracy of the deep-learning approach is the fact that 2D CNNs are unable to properly consider the temporal dimension that correlates subsequent frames in a video. The use of 3D CNNs could be an interesting way to pursue, and we plan to consider its inclusion in CAT-CAD. However, as highlighted in the paper, particular care should be taken to avoid overfitting, due to the fact that 3D networks require a much higher number of training samples than 2D networks, and this contrasts with the scarcity of quality-labeled videos. In this light, we advocate the construction of a benchmark for cataplexy detection which includes labeled video recordings of a large number of patients, although this is likely to raise issues regarding the privacy of patients (since videos displaying the patient face cannot be anonymized).

Another interesting research direction we are willing to pursue is the combination of the two detection techniques into a hybrid solution, able to exploit the best-of-both-worlds.

Finally, we plan to investigate, working conjointly with neurologists, the time correlation among the three motor phenomena. Indeed, in our current implementation, cataplexy is detected if any of the three symptoms is present, as confirmed by neurologists [32]. However, neurologists also suggested that the appearance of the three motor phenomena during a cataplexy crisis is sometimes sequential, with ptosis followed by mouth opening and (possibly) head-drop. Clearly, recognition of this motion behavior is a difficult task that involves complex time-correlated image analysis.

**Author Contributions:** Conceptualization, I.B.; methodology, I.B. and A.D.L.; software, A.D.L.; validation, I.B. and A.D.L.; formal analysis, I.B.; investigation, I.B. and A.D.L.; resources, I.B.; data curation, A.D.L.; writing—original draft preparation, I.B. and A.D.L.; writing—review and editing, I.B.; visualization, I.B. and A.D.L.; supervision, I.B.; project administration, I.B.; funding acquisition, I.B. All authors have read and agreed to the published version of the manuscript.

**Funding:** This research received no external funding.

**Institutional Review Board Statement:** Ethical review and approval were waived for this study, due to the fact that input video data were provided by the Outpatient Clinic for Narcolepsy of the University of Bologna. The use of such video data to diagnose cataplexy was originally approved by the local Ethic Committee [15].

**Informed Consent Statement:** Informed consent from all subjects involved in the experiments was obtained for the original study [15].

**Data Availability Statement:** The data presented in this study are available on request from the corresponding author. The data are not publicly available due to privacy.

**Acknowledgments:** The authors gratefully acknowledge the support of NVIDIA Corporation for the donation of the GPUs used for this research.

**Conflicts of Interest:** The authors declare no conflict of interest.

## Appendix A. Background

This appendix provides basic information on existing technologies that were used to implement the CAT-CAD tool, namely:

- the SHIATSU framework for video retrieval and analysis,
- the OpenFace library for detection of face landmarks,
- Convolutional Neural Networks (CNNs), which are the basic technology for the deep-learning approach, and
- the Tensorflow library, which was used to implement CNNs.

*Appendix A.1. The SHIATSU Framework*

SHIATSU is a general and extensible framework for content-based video retrieval [28,40]. Its engine consists of three main components:

1. The visual features extractor is used to automatically extract visual features from video frames, exploiting functionalities of the Windsurf library [41].
2. The annotation processor implements algorithms for the automatic tagging of videos by exploiting video features.
3. The query processor contains the logic for retrieval of videos based on semantics (tags) and/or similarity (features).

Specific services of SHIATSU used by CAT-CAD include:

- Splitting of videos into individual frames.
- Video playback.
- Persistence of video and metadata (labels manually entered by the neurologist or automatically extracted by the video analyzers) exploiting the MySQL RDBMS.
- Video retrieval based on tags: in particular, the multi-dimensional tagging approach already present in SHIATSU allows the coherent and seamless management of the 3-D tagging in CAT-CAD. This is because a single frame could be tagged to reflect the presence of any of the three independent motor phenomena.
- Feature-based video retrieval: since the feature used to assess similarity in CAT-CAD (percentage of presence of a specific motor phenomenon, see Section 3) is simpler than the visual features commonly exploited in SHIATSU, the query processor has been properly extended to exploit indexing functionalities of the underlying DBMS.

*Appendix A.2. OpenFace*

OpenFace (https://github.com/TadasBaltrusaitis/OpenFace, accessed on 30 March 2021) is arguably one of the most popular open-source library for facial landmark detection, able to localize facial features, such as eyes and eyelid contours, nose and mouth position, and so on [42]. OpenFace exploits a novel instance of the Constrained Local Model (CLM) framework called Constrained Local Neural Field (CLNF) dealing with the issues of feature detection in complex scenes reaching state-of-the-art performance when detecting facial landmarks across different illuminations. A CLM is a class of methods of locating sets of keypoints (constrained by a statistical shape model) on a target image. The general CLM approach consists of three basic steps:

1. Sample a region from the image around the current estimate.
2. For each keypoint, generate a *response image* giving a cost for having the point at each pixel.
3. Search for a combination of points optimizing the total cost, by manipulating the shape model parameters.

To perform the above steps, three main components are used:

- a Point Distribution Model (PDM),
- local detectors that evaluate the probability of a landmark being aligned at a particular pixel location, and
- the fitting strategy to be used.

The CLNF used by OpenFace is an instance of a CLM that uses more advanced local detectors and an optimization function (see [42] for details). The two basic components of CLNF are:

- a PDM which captures landmark shape variations, and
- local detectors which capture local appearance variations of each landmark.

*Appendix A.3. Convolutional Neural Networks*

Convolutional Neural Networks (*ConvNets* or *CNNs*) are a particular type of Artificial Neural Networks (*ANNs*) that have proven to be very effective when using images as input [43]: this makes them particularly suitable for image recognition and classification. One of the advantages offered by CNNs is the ability to use raw images as input, without the need to pre-process them, while one of the disadvantages is that they require very large sets of previously labeled images for training: this can be a problem in certain situations, such as ours for example, when the specific nature of the problem to be solved makes impossible to find public labeled datasets for network training. A common strategy, also used in this work, to get around this limitation is called *data augmentation*: it consists of the artificial enlargement of database by providing visual and label-preserving modifications to the available images.

Like ANNs, CNNs are organized into layers: in the following we will describe the most commonly used layers.

### Appendix A.3.1. Convolutional

Convolutional layers, as the name says, perform a convolution over the input. Let $f_k$ be the filter with a kernel size $n \times m$ applied to the input $x$, $k \in \mathbb{N}$ is the number of filters applied (the more filters are used, the more features are extracted from the image) and $n \times m$ is the number of input connections each CNN neuron has. The resulting output of the layer is calculated as follows:

$$C_k(x_{u,v}) = \sum_{i=-\frac{n}{2}}^{\frac{n}{2}} \sum_{j=-\frac{m}{2}}^{\frac{m}{2}} f_k(i,j) x_{u-i,v-j} \tag{A1}$$

The filters $f_k$ are realized by sharing weights of neighboring neurons: this has the positive effect that a lower number of weights must be trained, with respect to standard ANNs layers.

### Appendix A.3.2. Max Pooling

A Pooling layer takes a small rectangular block from the output of the previous layer and subsample it to produce a single output from that block, applying a function over the input $x_i$. In our architecture, we used Max Pooling layers: if $m$ denotes the size of the filter, the output of a Max Pooling layer is:

$$P(x_i) = \max_{k,l \in [-\frac{m}{2}, \frac{m}{2}]} \left\{ x_{i+k,i+l} \right\} \tag{A2}$$

This layer features translational invariance with respect to the filter size.

### Appendix A.3.3. Rectified Linear Unit

A Rectified Linear Unit (ReLU) is an element-wise operation (applied per pixel) that replaces all negative pixel values with zero, using the following activation function to calculate its output given $x$:

$$R(x) = max(0; x) \tag{A3}$$

The purpose of ReLU is to introduce non-linearity in CNNs.

Appendix A.3.4. Local Response Normalization

A Local Response Normalization layer (LRN) is used to normalize the output of a previous layer before sending it to non-linear layers, such as ReLU. The goal of this type of layer is to bring inputs to the following layer to a common scale. Since it is important to detect high frequency features with a large response, sensitivity of an excited neuron is increased, with respect to its neighbors, if values are normalized around its local neighborhood.

Appendix A.3.5. Fully Connected

In a Fully Connected layer all neurons are connected to all neurons of the prior layer. Let the input be $x$ with size $k$ and $l$ be the number of neurons in the fully connected layer, this results in a matrix $W_{l \times k}$, so the output of the layer will be:

$$F(x) = \sigma(W * x) \tag{A4}$$

where $\sigma$ is called *activation function*.

Appendix A.3.6. Output

The Output layer produces the result in the form of a so-called *one hot vector*. A one hot vector is a vector with the dimensionality of the number of classes, where all values are 0 except for the predicted class, for which the value is 1. The resulting class $C(x)$ for the output vector $x$ is:

$$C(x) = \arg \max_i \{x_i\} \tag{A5}$$

*Appendix A.4. Tensorflow*

TensorFlow (https://www.tensorflow.org/, accessed on 30 March 2021) is an open-source and cross-platform software library for numerical computation using dataflow graphs [44]. Every node of the graph is a mathematical operation, while edges connecting nodes are multi-dimensional data arrays (the name TensorFlow derives from these arrays, which are called "tensors"). The entire dataflow graph is a thus complete description of computations, which occur within a session, and can be executed on devices, such as CPUs or GPUs.

TensorFlow was originally developed by the *Google Brain* team for research on machine learning and deep neural networks, but the generality of the system led to its exploitation for a wide range of tasks. One of the key features of TensorFlow is that it can move execution of the computationally expensive tasks of a given graph from solely CPU to heterogeneous GPU-accelerated environments, without significant alteration of code. Originally developed for Google internal use only, it was released under the Apache 2.0 open-source license in 2015, with the aim to bring parallelism and high scalability to machine learning for everyone.

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
