# Peer review of "CAT-CAD: A Computer-Aided Diagnosis Tool for Cataplexy"

_computers, doi:10.3390/computers10040051_

Round 1
Reviewer 1 Report
It is a tedious process to manually analyze and annotate videos, especially for real time videos. An automatic tool developed based on available open-source tools could benefit the clinical workflow and throughput for patients and clinicians. The aim of this work was to develop an automatic computer-aided diagnosis tool for cataplexy using two approaches: pattern-based and deep learning-based, and to compare the performance between these two approaches. It is very valuable and practical to have an automatic tool for neurologists to diagnose cataplexy; however, the motivation of developing deep learning-based tool is not clear as its performance is yet to be comparable with the pattern-based one. There are some comments on the deep learning-based approach and modifications might be needed.
Comments
Page 11, line 394: Have the authors tried to put three motor phenomena into only one neural network for the classification? If so, how was the performance compared to that with three combined, and would it be better to avoid the overall performance degradation mentioned by the authors in the manuscript?
Page 12, Table 1: In the Input Block row, why the image dimensions were changed from 80 to 56 as the output of LRN1 layer?
Page 12, Table 1: In Output Block row, the layer naming as Classifier might not be entirely correct because the outputs of the final layer (fully connected layer) are probabilities and must undergo classification using a threshold. Or please describe what the 2 x 1 x 1 outputs are and add the missing operation/layer if there is any.
Page 12, line 423: Given the information from Figure 7, the image dimensions should not change before and after the first convolutional layer. Please verify and revise if necessary.
Page 13, Figure 6 and 7: Please add the number of channels to each layer to increase the clarity. In addition, please use the same names of the layers in Table 1 and Figure 7.
Page 16, line 528: Although a fn can be corrected by tp from the other two motor phenomena, fp cannot be corrected by tn. Please explain why fp can be transformed into a tp? In addition, please refer to the first comment on page 11, line 394, to see if combining three phenomena into one neural network can help.
Page 20, line 631: Was the range of rotation angle from 0º to 90º reasonable to augment the data? Would +/- 45º be more reasonable and improve the training performance?
Page 22, line 704: Since the deep learning-based approach is not yet comparable to the pattern-based approach, could the authors revise the motivation and objective to clarify that providing a pattern-based approach for neurologists is first aim of this work and in the meantime the deep learning-based is still under investigation for performance improvement? It could avoid unnecessary confusion.
Reviewer 2 Report
The paper is well organized with proper structure and length. The bibliography is sufficient and well given.
Specifically, the technical terms are explained in detail and the topic of the paper is clear and understandable.
Mathematical models are well written and appropriate refereed
The presented methodology and the results are clearly communicated, with the necessary background for the readers included in the paper.
The review of the state-of-the-art is sufficient. It includes lots of references to other relevant studies that have been previously proposed for the discovery of relations.
The novel contribution of the paper is highlighted, as well.
The conclusion section includes a discussion about the results obtained by this work and the previous works on the analysis of the same or similar data.
Reviewer 3 Report
In this article, the authors present the CAT-CAD tool for the automatic detection of cataplexy.
First of all, the article is well written and well organized. Minor details on English grammar require review.
The authors have presented a detailed analysis of the proposed tool, and the presentation is solid.
The introduction provides sufficient background. The technical contribution is high and clear.
The authors have described the advantages and limitations of the proposed approach. They have demonstrated the environment of the experiment in detail.
There is a clear presentation of the results and their commentary.
I consider that the article has excellent potential. However, before being considered ready for publication, some aspects need to be clarified and improved.
1)My suggestion is to increase the number of references to allow for deeper discussion and comparisons with the state-of-the-art techniques.
2)I recommend the authors include more recent references and from MDPI as well.
3)In the Discussion section, I suggest the authors provide information about the cognitive features of narcolepsy with cataplexy embedded in the proposed system. How CAT-CAD contributes to its diagnosis? Also, in what other disorders could this tool contribute to their diagnosis?
To sum, it is clear the overall merit and contribution of this work. The presentation is sufficient for a research article.
